# The Influence of UV Radiation Aging on Degradation of Shear Thickening Fluids

**DOI:** 10.3390/ma15093269

**Published:** 2022-05-02

**Authors:** Radosław Żurowski, Mariusz Tryznowski, Selim Gürgen, Mikołaj Szafran, Aleksandra Świderska

**Affiliations:** 1Faculty of Chemistry, Warsaw University of Technology, Noakowskiego 3, 00-664 Warsaw, Poland; rzurowski@ch.pw.edu.pl (R.Ż.); szafran@ch.pw.edu.pl (M.S.); aswiderska23@wp.pl (A.Ś.); 2Faculty of Mechanical and Industrial Engineering, Warsaw University of Technology, Narbutta 85, 02-524 Warsaw, Poland; 3Department of Aeronautical Engineering, Eskişehir Osmangazi University, 26040 Eskişehir, Turkey; sgurgen@ogu.edu.tr

**Keywords:** shear thickening fluids, aging, lightfastness, product degradation

## Abstract

Shear thickening fluids (STFs) are innovative materials that can find applications in smart body armor. However, the usage of STFs is limited by the aging of these materials. This work aims to analyze the influence of UV radiation on the aging process of STFs. The investigation was done experimentally, and artificial aging was applied to investigate the impact of UV radiation on the properties of STFs. The shear-thickening properties of obtained STFs were confirmed by viscosity measurements. The STFs based on PPG425, PPG2700, and KE-P10 exhibited a very high maximum viscosity of up to 580.7 Pa·s and 3313 Pa·s for the STF425 and STF2700, respectively. The aging of the obtained STFs caused the liquid matrix degradation, causing damage to the STFs and their change from liquid into solid. Furthermore, the FT-IR, ^1^H NMR, and ^13^C NMR spectroscopies were used for the confirmation of the breakdown of STFs. The FT-IR spectroscopy revealed the appearance of carbonyl groups in STFs after aging. Moreover, ^1^H NMR and ^13^C NMR spectroscopy confirmed the formation of the typical groups containing carbonyl groups. Our results demonstrate that STFs are UV light-sensitive and may lose their properties during storage.

## 1. Introduction

The need for innovative and intelligent materials is encouraging the search for and development of novel materials and applications. An example of such materials is shear thickening fluids (STFs). These are non-Newtonian fluids, and their viscosity increases with either increasing shear rate or applied stress [1]. STFs are ceramic-polymer composites, wherein a ceramic powder is dispersed in an organic matrix such as glycerin, poly(ethylene oxide), or poly(propylene glycol). These are technically not new materials; STFs have been known for many years [2]. One of the material properties of STFs is the logarithmic increase of viscosity as a function of shear rate, known as the dilatation jump and observed as the liquid-to-solid transition. The viscosity of STFs can increase rapidly, and they can appear as solid-like suspensions. This behavior may be utilized in many applications, such as in body armor or various other elements of human protection [3].

The STFs can be used to impregnate the ballistic fabrics, improving the fiber fabric’s friction and energy absorption [3]. For this purpose, high-performance fabrics can be used, for example Kevlar^®^ [4,5] or UHMWPE (Ultra High Molecular Weight Polyethylene) [6]. Arora et al. have reported the usage of panels made from up to five layers impregnated with STFs [5]. In this work, it was discovered that the angular orientation of fabrics highlights the synergic effect of impregnating fabrics with STFs. Other researchers proposed hybrid composite materials and novel composite materials based on natural fibers and cornstarch with STFs layers [7]. Finally, aramid fabrics may be impregnated with STFs and combined with a thermoplastic polyurethane coating, resulting in sandwich-structured composite panels [8]. Chang et al. have reported the usage of polyurea elastomer/Kevlar fabric composites enriched with STFs as protective materials [9]. Despite of the type of STFs, the studies on the topic of STFs confirm the bullet resistance of Kevlar^®^ fabrics reinforced with STFs [6,7]. Unfortunately, the resistance of fabrics impregnated with STFs declines when the fabrics contain too many STFs [4]. Aside from the application of STFs in human protection, other applications have been reported in the literature. For example, Gürgen et al. reported using STFs in cutting tools to prevent vibration dumping [10]. Additionally, Liu et al. proposed using STFs with magneto-sensitive properties as shock absorbers for high-end vehicles [11]. Furthermore, STFs can be used in multilayer composites for energy-absorbing purposes [12].

STFs can be composed of various liquid matrices, which include PEGs, PPGs, and even ionic liquids. The most popular materials for the STFs preparation, due to their good properties and low price, are various PPGs. In our research team, STFs were prepared from PPGs characterized by different molecular masses and silica (see for example in [13,14,15]). Qin et al. have proposed novel STFs based on various imidazolium and pyridinium ILs characterized by a maximum viscosity of up to 1000 Pa·s [16].

There are some disadvantages of introducing STFs into mass-scale use. Despite several literature reports, the effect of shear thickening is not fully understood nor clearly defined. However, several researchers have presented modeling of the shear thickening effect in STFs [17,18,19,20,21]. Furthermore, STFs, like other materials, exhibit certain stability and lifetime. The stability and shear thickening properties of STFs strongly limit the manufacturing of products such as protective devices [22]. According to the literature, STFs seem to exhibit excellent stability [13]. However, the conditions of storage and usage may influence the lifetime of the material. For example, the presence of water in the environment where STFs are stored decreases their viscosity [23]. Finally, there is a technological barrier to obtaining STFs in large amounts for mass-scale adoption.

The application of STFs is strongly limited by their lifetime and the nature of their usage and storage before usage. Apart from the aging of the STFs, the aging of fabrics in the body armor can be observed [24]. Those two effects strongly limit the expiration date of body armor. To the best of our knowledge, the aging of STFs has not been previously reported in the literature. Nakonieczna et al. have reported excellent stability after 8 days of STFs based on PPG and amorphous silica [13]. Furthermore, Żurawski et al. have compared the rheological behaviors of STFs after different periods of time and discovered that STFs demonstrated no changes in dilatant effect [25]. This work aims to analyze the influence of UV radiation on the aging process of STFs. The STFs can be stored prior to usage in various conditions, whereas they can be exposed to external factors, for example light. Additionally, the high-performance fabric panels used in body armors have a warranty period and use-by date (approximately 5–10 years depending on the producer), and after that they should not be used. The method and intensity of bulletproof vest usage also affects the quality of the panels based on high-performance fabrics. The aging with UV light gives an overview of the stability of STFs and their shelf life, especially when the accelerated aging process is used. For this work, STFs with high maximum viscosity were prepared. Next, the aging process was performed on the obtained samples. Comprehensive analyses of STF aging products were performed, including FT-IR, ^1^H NMR, and ^13^C NMR spectroscopies. For the first time, we demonstrate the limitations of the wide usage of innovative and smart materials like STFs.

## 2. Materials and Methods

### 2.1. Materials

For STF preparation, poly(propylene glycol) PPG425 and PPG2700 (CAS 25322-69-4, Sigma-Aldrich, St. Louis, MO, USA) and spherical silica powder KE-P10 (Nippon Shokubai, Tokyo, Japan), with a particle size within the range of 100–200 nm, specific surface area of 132.2 m^2^ g^−1^, and density of 1.96 g cm^−3^ [26], were used. The properties of poly(propylene glycol) types used in this work are summarized in Table 1.

### 2.2. STF Preparation

The STFs were produced by mixing poly(propylene glycol) with silica powder. The Computer Aided Design, CAD, scheme of the mixer is displayed in Figure 1. The laboratory stand was specially built for the STF preparation. The mixer is equipped with a glass 250 mL reactor and a mechanical stirrer (R50D, Ingenieurbüro CAT, Ballrechten-Gottingen, Germany) equipped with a stainless-steel propeller-mixing geometry. The silica powder was added stepwise. The applied mixing speed was within the range of 150–200 rpm. The solid-phase concentration of the fluid was 50 vol. %. Table 2 displays the composition of STFs.

### 2.3. Rheology Measurements

The rheological behavior of STFs was studied using a rational KinexusPro rheometer (Malvern Panalytical, Malvern, Worcestershire, UK) with parallel plate geometry (top plate *φ* 20 mm; bottom plate *φ* 100 mm; spacing gap 0.7 mm) with applied shear stress in the range of 10^−2^ up to 10^3^ s^−1^. The measurements were repeated twice with fresh samples.

### 2.4. Artificial Aging

Artificial aging was carried out using a Suntest CPS+ (Atlas, Mount Prospect, IL, USA). The device was equipped with an optical filter which is a UV external light filter that eliminates UV radiation with wavelengths shorter than 290 nm. The applied light intensity was 580 W m^−2^. The Black Standard Thermometer (BST) was set at 35 °C. A small amount (approximately 10 g) of each STF was placed in a glass Petri dish (*φ* 80 mm) covered with a quartz lid. The samples subjected to artificial aging are displayed in Figure 2. Blue Wool Scale samples (approximately 1 cm^2^) were exposed together with STF samples. The contrast between exposed and unexposed Blue Wool Scales stripes was measured after the aging cycle using greyscale, according to ISO 105-A02:1993 standard. The procedure is described elsewhere [27]. The tests were performed for a total of 167 h, which corresponds to approximately 15.5 weeks (108 days) of natural aging. Every 1 h spent under aging conditions corresponds to a 1980 kJ m^−2^ dose absorbed by the samples.

### 2.5. Product Analyses

FT-IR, ^1^H NMR, and ^13^C NMR spectroscopies were used to characterize the emerging products during aging. The analyses were performed both prior to and after aging.

The attenuated total reflectance ATR FT-IR spectra were recorded within the wavelength range of 400–4000 cm^−^^1^, with a resolution of 4 cm^−^^1^ using a Nicolet iS5 (Thermo Fisher Scientific, Waltham, MA, USA) spectrometer. The FT-IR spectra were analyzed with OMNIC Specta™ software (series 9.12.968, Thermo Fisher Scientific, Waltham, MA, USA).

^1^H NMR and ^13^C NMR spectra were recorded using a Varian VXR 400 MHz spectrometer (Varian, Palo Alto, CA, USA) with tetramethylsilane as an internal standard and deuterated solvents (CDCl_3_). The results were analyzed with MestReNova v.6.2.0 (Mestrelab Research S.L, Santiago de Compostela, Spain) software.

## 3. Results and Discussion

### 3.1. Flow Fluid Properties

Figure 3 demonstrates the viscosity curves of the STFs. The results confirm the typical shear thickening behavior of the obtained STFs. The maximum viscosity of STF2700 is five times higher than for STF425. The STF2700 exhibited a dilatant effect at 4.4 s^−1^ with a maximum viscosity of 3313 Pa·s, and STF425 exhibited a dilatant effect at 12.7 s^−1^ with a maximum viscosity of 580.7 Pa·s.

The dilatant effect observed for STF2700 is much higher than for those previously reported in the literature. For example, Głuszek et al. reported the maximum viscosity for STFs based on KE-P10 and poly(propylene glycol) (molar mass 1000 g mol^−1^) to be above 350 Pa·s [26]. In this work, the maximum viscosity reaches up to 1000 Pa·s [5,16,26,28]. The maximum viscosities reported by Arora et al. and Bajya et al. exceeded 170 Pa·s [5,28]. However, Qin et al. reported STFs with higher solid contents than STFs obtained by us, with maximum viscosities above 800 Pa·s [4]. Furthermore, modification of carrier liquid does not reflect much higher values of maximum viscosity. For example, Ghosh et al. have reported a simple modification of polyglycols with citric acid, and maximum viscosity increased 76 times with the peak viscosity up to 800 Pa·s [29].

### 3.2. Influence of Artificial Aging

The total aging time together with an absorbed dosage of 1980 kJ m^−2^ h^−1^ corresponds to about approximately 15.5 weeks under natural conditions. Additionally, the fading of Blue Wool Scale #5 was observed, corresponding to approximately three months of exposure to light under natural conditions [30].

The samples after aging are displayed in Figure 4. The total degradation of STFs was observed, hence the product analyses (FT-IR, ^1^HNMR, and ^13^CNMR) were performed without viscosity measurements. The condition of the STF samples after aging enables the viscosity measurement. During the aging of STFs, the change from liquid- to solid-state was observed. It is related to the degradation and breakdowns of the PPGs macromolecules, which are liquid-phase forming a solvation layer on the surface of the silica granules. Therefore, depending on the degree of PPG degradation during aging, liquid STFs changed into solid-state. The product analyses with FT-IR, ^1^HNMR, and ^13^CNMR confirmed the degradation of the PPGs during aging.

The FT-IR spectra of STF2700 before and after aging are displayed in Figure 5. The FT-IR spectra were measured with the ATR technique and transferred to KBr using Omnic software. In the FT-IR spectra, characteristic changes within the range of 3000–3600 cm^−1^ and 1700–1800 cm^−1^ are observed after aging. Peak broadening within the range of 3000–3600 cm^−1^ is related to the presence of –OH groups, which appear with the degradation of the carrier fluid into macromolecules with shorter chains. Finally, a new peak appears at 1720 cm^−1^, related to the presence of carbonyl groups (–C=O).

PPG2700 and PPG425 were analyzed after exposure to UV radiation, using ^1^H NMR and ^13^C NMR spectroscopies to identify the degradation products.

In the ^1^H NMR spectrum of raw PPG2700 there are peaks between 3–4 ppm corresponding to the chemical shift range of methine and methylene groups bound to the oxygen atom, and a broadened peak at 1.1 ppm corresponding to methyl groups. The integrals of both signals are equal. In the ^13^C NMR spectrum there are peaks between 71–77 ppm corresponding to methine and methylene groups; peaks at 66.9, 65.3, and 65.0 ppm corresponding to CH_x_–OH; and peaks between 17–18 ppm corresponding to methyl groups. Additional signals appeared in the ^1^H and ^13^C NMR spectra of PPG2700 after UV exposure. Analysis and interpretation of the spectra displayed that the new peaks are characteristic of the formic ester and other esters, consistent with the results presented in previous reports on the thermal degradation of PEG or PPG as a result of oxidation [31,32,33,34,35]. Peaks appeared in the ^13^C NMR spectrum at 207.4, 200.0, 172.8, 170.4, 162.0, and 160.5 ppm, characteristic of carbonyl atoms from ketones, aldehydes, and esters, including formic esters, respectively. In the ^1^H NMR spectrum, signals appeared that are characteristic of aldehydes (9.6 ppm) and formic esters (7.9 ppm). Additionally, signals between 3.5–5.0 ppm and 1.9–2.1 ppm verify the formation of esters. The assignment of signals to their appropriate groups is displayed in Table 3, and in Figure 6 and Figure 7.

## 4. Conclusions

For the first time, the influence of UV exposure time on the properties of STFs was measured after artificial aging. The STFs obtained in this work exhibited a shear thickening effect up to a maximum of 3313 Pa·s, exhibiting a much more significant shear thickening effect than previously observed in the literature.

The aging time of STF samples corresponds to approximately three months of natural aging indoors, and the aging causes visible changes in their appearance. FT-IR spectroscopy and ^1^H NMR and ^13^C NMR spectroscopies confirm the degradation of the obtained STFs. The spectrometric analyses confirm the appearance of carbonyl groups during aging. Therefore, the degradation causes the shortening of the oligomer macromolecule chains, related to the change of STFs during UV aging from a liquid into a solid, making it impossible to perform repeated rheological tests.

Our results reveal the necessity of preparation and storage of fluids under reduced UV radiation conditions to prevent the degradation of STFs and the significant deterioration of their properties.

Our results can be summarized in a few points:We have obtained STFs with a very high dilatant effect (the shear thickening effect of up to a maximum of 3313 Pa·s) not previously reported in the literature.The UV radiation causes the damage of the STFs and the change of the STFs from the liquid to solid.The results indicate the need for appropriate storage of STFs in the UV-free environment prior to and during usage.

## Figures and Tables

**Figure 1 materials-15-03269-f001:**
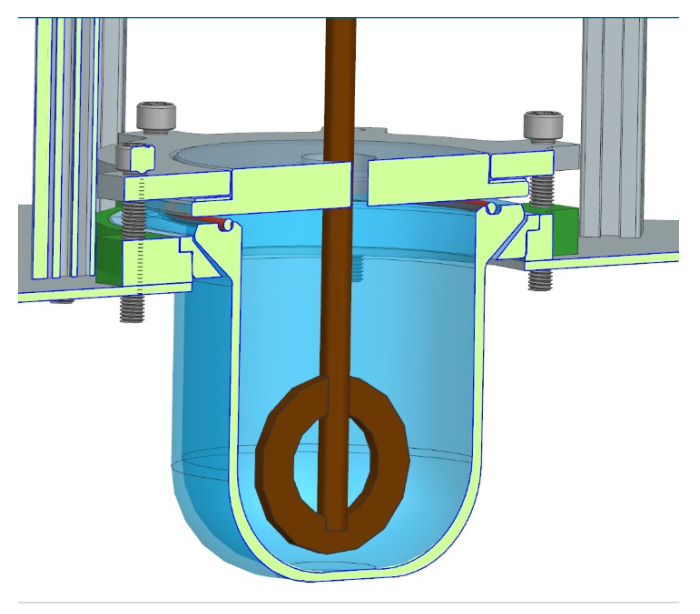
Scheme of the mixer used for the STF preparation.

**Figure 2 materials-15-03269-f002:**
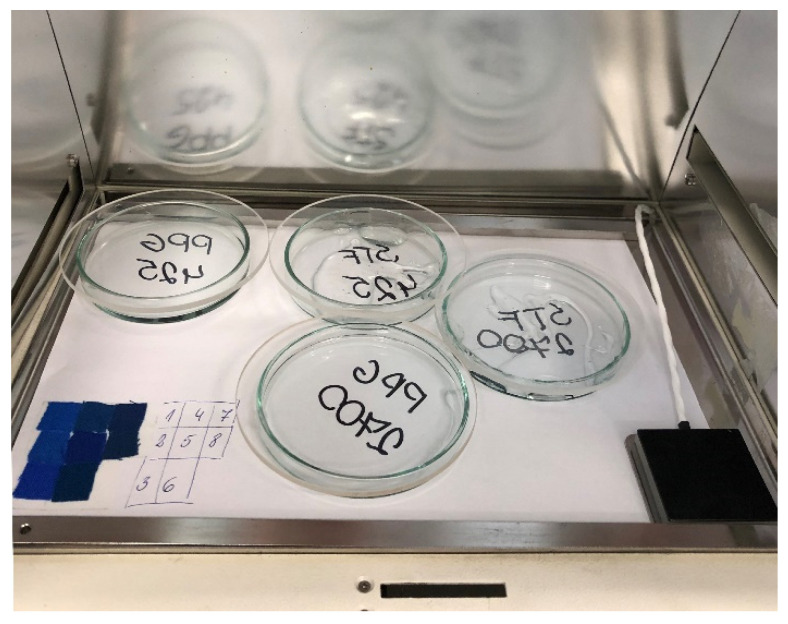
PPG and STF samples together with Blue Wool Scale prepared for aging.

**Figure 3 materials-15-03269-f003:**
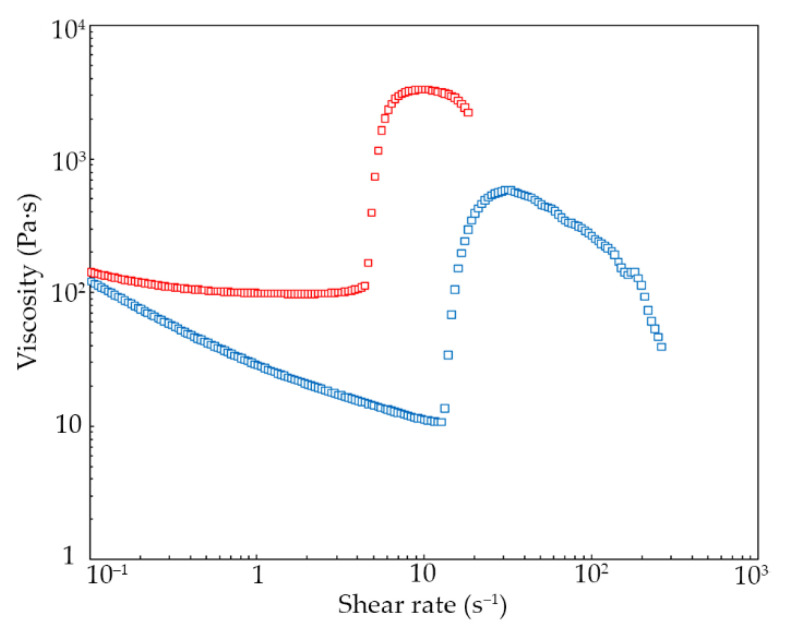
Viscosity vs. shear rate for the investigated STFs: blue squares STF425, red squares STF2700.

**Figure 4 materials-15-03269-f004:**
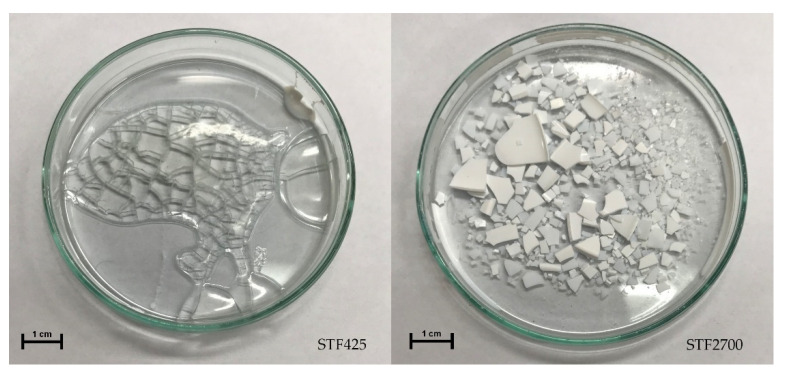
Samples of STFs after aging.

**Figure 5 materials-15-03269-f005:**
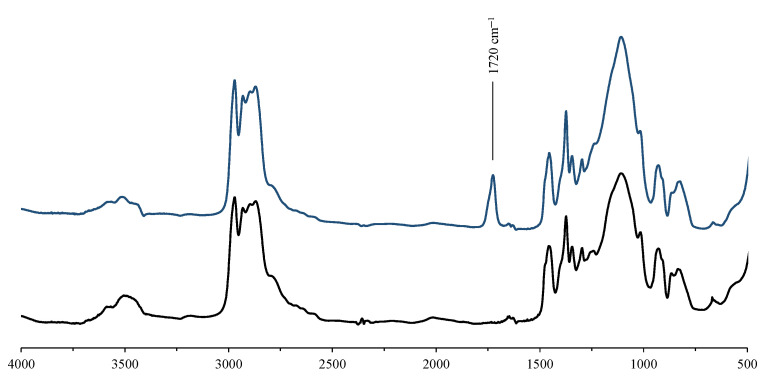
FT-IR spectra of STF2700 before (black line) and after (dark blue line) aging.

**Figure 6 materials-15-03269-f006:**
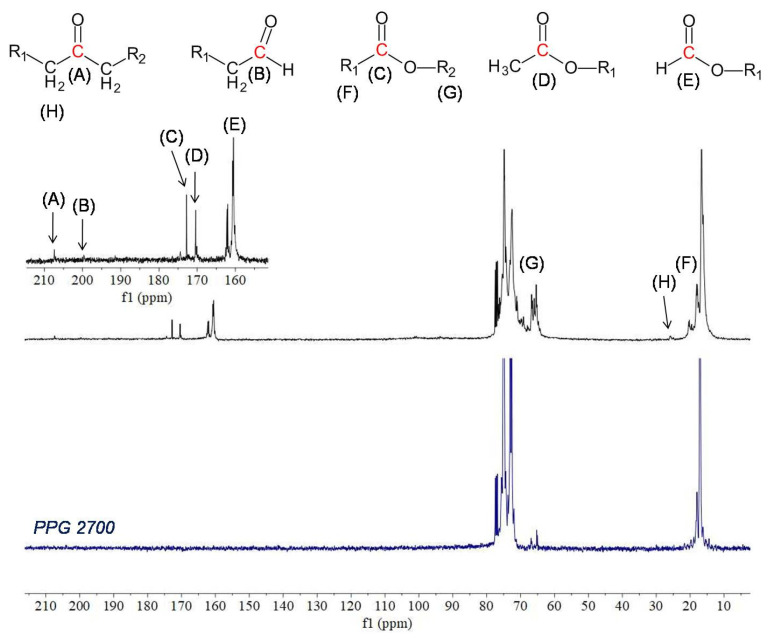
^13^C NMR spectra (CDCl_3_, 100 MHz) of raw PPG2700 (blue) and PPG2700 exposed to UV radiation (black). Inset: enlarged region within the 150–210 ppm spectral range after UV exposure.

**Figure 7 materials-15-03269-f007:**
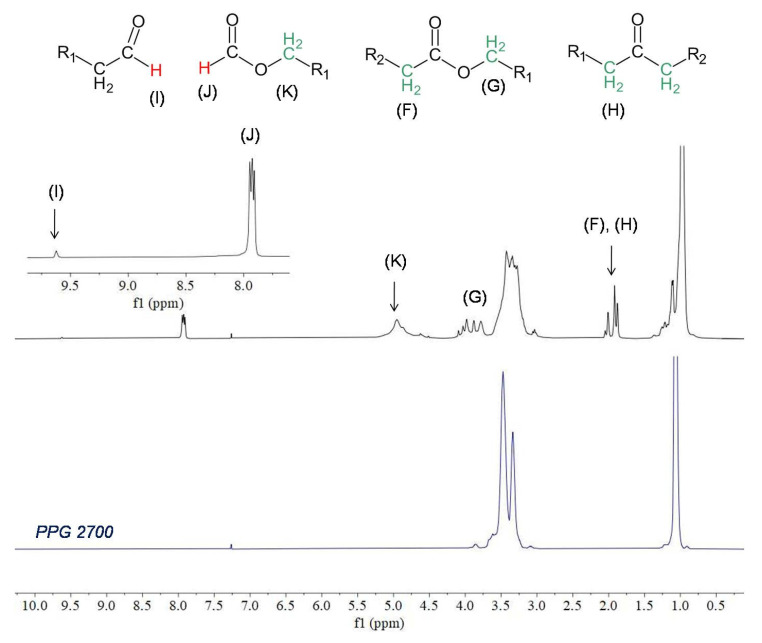
^1^H NMR spectra (CDCl_3_, 400 MHz) of raw PPG2700 (blue) and PPG2700 exposed to UV radiation (black). Inset: enlarged region within the 7.5–10 ppm spectral range after UV exposure.

**Table 1 materials-15-03269-t001:** Properties of poly(propylene glycol) ^1^.

Abbreviation	*M*_n_ (g mol^−1^)	Density (g cm^−3^)
PPG425	~425	1.01
PPG2700	~2700	1.01

^1^ According to the Safety Data Sheet provided by the supplier.

**Table 2 materials-15-03269-t002:** Properties of spherical silica powders.

Abbreviation	Ceramic Powder	Carrier Fluid
STF425	KE-P10	PPG425
STF2700		PPG2700

**Table 3 materials-15-03269-t003:** Chemical structures and assignment of characteristic groups to signals in ^1^H and ^13^C NMR spectra.

Chemical Structure	Chemical Shift (^1^H NMR)	Chemical Shift (^13^C NMR)
** 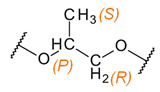 **	*P*, *R*: 3–4 ppm *S*: 1.1 ppm	*P*, *R*: 71–77 ppm *S*: 17–18 ppm
** 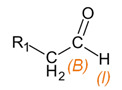 **	*I*: 9.6 ppm	*B*: 200.0 ppm
** 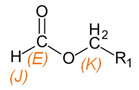 **	*J*: 7.9 ppm *K*: 5.0 ppm	*E*: 160–162 ppm
** 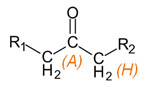 **	*H*: 1.8–2.2 ppm	*A*: 207.4 ppm *H*: 25.8 ppm
** 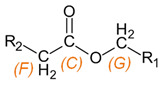 **	*G*: 3.7–4.3 ppm *F*: 1.8–2.2 ppm	*C*: 170–173 ppm *G*: 63.5–68.0 ppm *F*: 18–20 ppm

## Data Availability

The data presented in this study are available on reasonable request from the corresponding author.

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
