# Peer review of "The Influence of UV Radiation Aging on Degradation of Shear Thickening Fluids"

_materials, 2022, doi:10.3390/ma15093269_

Round 1
Reviewer 1 Report
Thank you for submitting your paper. The work done here draws attention to a significant subject shear thickening fluids and UV radiation in composites. I have found the paper to be interesting. However, several issues need to be addressed properly before the paper is being considered for publication. My comments including major and minor concerns are given below:
- Please consider reviewing the abstract and highlight the novelty, major findings, and conclusions. I suggest reorganizing the abstract, highlighting the novelties introduced. The abstract should contain answers to the following questions:
- What problem was studied and why is it important?
- What methods were used?
- What conclusions can be drawn from the results? (Please provide specific results and not generic ones).
- The abstract must be improved. It should be expanded. Please use numbers or % terms to clearly shows us the results in your experimental work.
- Please consider reporting on studies related to your work from mdpi journals.
- The introduction must be expanded, please consider improving the introduction, provide more in-depth critical review about past studies similar to your work, mention what they did and what were their main findings then highlight how does your current study brings new difference to the field.
- The title of the manuscript does not read well, the authors should consider improving it.
- In line 72 the authors cite reference number 24, but when checking the paper, it is not clear why is this reference used here?
- The authors should add more graphs and images in the materials and methods section. After all, this is an experimental study and images of equipment used, fabricated samples and pre and post tested ones should be shown.
- In figure 2, how can the authors correlate the density of the two fluids to the effects of viscosity? Do the two fluids have same density?
- In line 119 when the authors say more signficant they should clearly indicate by how much? Use % or numbers to report that. Authors need to do this in other parts of the paper where they report similar type of results.
- Line 135 why you have one single line alone there? The authors need to combine all small paragraphs into larger ones. Anything less than 4-5 lines should be combined with previous or following paragraph.
- Figure 3 add a scale bar to the petri dishes.
- Where is the discussion section? I can only see section called Results?
- The results are merely described and is limited to comparing the experimental observation and describing results. The authors are encouraged to include a more detailed results and discussion section and critically discuss the observations from this investigation with existing literature.
- Conclusion can be expanded or perhaps consider using bullet points (1-2 bullet points) from each of the subsections.
Author Response
We would like to thank the Reviewer for all the valuable comments. The response is in a separate list of changes. Please see the attachment.

Reviewer 2 Report
In the manuscript, authors have analyzed the influence of UV radiation on the aging process of STFs. The experimental study and artificial aging method were applied to investigate the impact of UV radiation on properties of STFs, later on validated by viscosity measurements. Results reveal that STFs are light-sensitive and lose their properties during storage.
In general results were somewhat predictable without running NMR and FTIR. UV will visibly damage polymer layers ( PPG , which is alcohol derivative monomer to start with). It will get oxidized in 15 weeks ( that is understandable).
But I wanted to see
- what happened to silica powder ( KE-P10)?
- Did it reacted with PPG?
- Did they run negative and positive control ?
- Why Only Polypropylene glycol was selected ?
- Why other classes of polymers were not included for comparative study?
- UV was going to deteriorate most of the polymers but why not find which class performed better?
May be they will include those studies in future research.
Anyway, manuscript is well written and results showing oxidation of PPG as I would have expected. So, could be published in the present form after addressing the above mentioned observations.
Author Response

(The authors gave the same response as above.)

Reviewer 3 Report
The manuscript "Investigation into the Aging of Shear Thickening Composites by UV Radiation: Product Analyses" presents a study on composites of polypropylene glycol and silica powder that are then characterized in terms of (a) shear thickening, (b) and chemical aging by FT-IR, 13C-NMR and 1H-NMR under light - including UV - exposition. A photo of the aged samples was taken to demonstrate the brittleness. Prior to the analysis the preparation of the two different samples with 2 molar masses of the polymer was described.
The whole presentation is straight forward and can be understood well. The chemical decay could be commented a little more. However, the selected substances are of more academic interest and would not be selected for the proposed purposes - like armor. So in my view the paper presents a way of analyzing such materials rather than giving deeper insight in most suitable materials.
Since - all in all - the paper is presented clearly, I would plead for publication after minor changes.
So my main point is that the chemical decay could be discussed a little more. No hint was given to the brittleness (except the two sentences for the photo and in the summary naming the aged samples solid). Also the spectra could highlight specific lines on order to argue for the solidification. The reader is left alone with drawing his own conclusions from the measurements.
Author Response

(The authors gave the same response as above.)

Round 2
Reviewer 1 Report
The authors provide superficial answers and changes made to the questions raised to previous round. please read each question carefully and improve the introduction and abstract.
Introduction is short and should be expanded.
Bulk citations are not allowed and its an abuse of citations to add 9 references in line 39 for one sentene.
Check line 198
The paper suffers from lack of critical discussion, only explaining what we can see from the trends in the graphs
Line 160 "break downs" not breaks down.
